# Increased Risk of Sjögren’s Syndrome in Patients with Obsessive-Compulsive Disorder: A Nationwide Population-Based Cohort Study

**DOI:** 10.3390/ijerph18115936

**Published:** 2021-06-01

**Authors:** Yi-Jung Chang, Jui-Cheng Tseng, Pui-Ying Leong, Yu-Hsun Wang, James Cheng-Chung Wei

**Affiliations:** 1Department of Pediatrics, Chang Gung Memorial Hospital, Taoyuan 33305, Taiwan; r64321@gmail.com; 2College of Medicine, Chang Gung University, Taoyuan 33302, Taiwan; 3Division of Allergy, Immunology and Rheumatology, Department of Medicine, Kaohsiung Veterans General Hospital, Kaohsiung 81362, Taiwan; jctseng@vghks.gov.tw or; 4Division of Allergy, Immunology and Rheumatology, Department of Medicine, Chung Shan Medical University Hospital, Taichung 40201, Taiwan; fiona.leong@gmail.com; 5Institute of Medicine, Chung Shan Medical University, Taichung 40201, Taiwan; 6Department of Medical Research, Chung Shan Medical University Hospital, Taichung 40201, Taiwan; cshe731@csh.org.tw; 7Graduate Institute of Integrated Medicine, China Medical University, Taichung 404333, Taiwan

**Keywords:** obsessive-compulsive disorder, Sjögren’s syndrome, cohort study, NHIRD, database

## Abstract

Obsessive-compulsive disorder (OCD) includes a wide range of symptoms and is often associated with comorbidities. Although psychiatric involvement may be an early manifestation of Sjögren’s syndrome (SS), only a few studies have demonstrated the relationship between OCD and SS. This is a nationwide cohort study identifying the risk of SS in OCD patients. We studied a longitudinal health insurance database for the period from 1999 to 2013. The study group was OCD patients with at least three outpatient visits or one hospitalization. The comparison cohort was matched by age and sex, as well as comorbidities. We calculated the risk of Sjögren’s syndrome using Cox proportional hazard regression models. We performed a propensity score match for confounders and effect modifiers between the two groups. The propensity score probability was estimated through logistic regression. Primary outcome was the incidental SS. A total of 1678 patients with OCD (49% women, mean age: 35.6 years) and 3356 controls were followed up, resulting in 13,077 and 25,856 person-years, respectively. The hazard ratio for developing SS was 3.31 (95% C.I.: 1.74–6.28) in patients with OCD, compared to those without OCD after adjusting for age, sex, and comorbidities. Furthermore, the risk of SS significantly increased over the 2-year follow-up period after OCD diagnosis. We concluded that risk of SS is significantly increased in patients with OCD compared to those without OCD. Clinically, Sjögren’s symptoms in OCD patients should be regularly assessed.

## 1. Introduction

Sjögren’s syndrome (SS) is one of the most common autoimmune diseases occurring worldwide, and its prevalence is approximately 0.5 to 1.5% of the general population [1,2]. Patients with SS are characterized by a broad spectrum of glandular and extraglandular features [1]. The extraglandular manifestations include focal neurologic deficits, diffuse cerebral involvement, and psychiatric issues. Psychiatric disorders such as anxiety, psychosis, cognitive impairment, and mood disorders are common comorbidities in patients with SS [3,4,5]. Psychiatric disorders are related to outcomes and concerns in autoimmune diseases. Furthermore, there is an increasing awareness that psychiatric manifestations can be initial symptoms of SS [6,7].

Psychological abnormalities usually accompany chronic diseases. As previous studies have revealed, neuropsychiatric manifestations in SS can occur during its course as well as at the onset of the disease. Obsessive-compulsive disorder (OCD) may present as an initial symptom and precede the well-recognized typical form of SS [6,7,8,9,10]. Key expressions include chronic, recurrent, and uncontrollable thoughts and behaviors. OCD is a universal psychiatric disorder and a significant health-economic burden with a prevalence of 2.5–3% in the general population [11,12]. Over the last few years, studies have reported an increased prevalence of OCD in patients with autoimmune diseases up to 17.8% [13], which may disturb their comprehensive health and lead to a higher degree of distress [14,15].

Therefore, for the prevention and early detection of SS, investigations into the impact of OCD on SS have become a critical. However, studies regarding the relationship between OCD and SS are very scarce and primarily consist of case reports. Therefore, the impact of OCD on SS remains unclear. To address these gaps, we conducted a nationwide, longitudinal, population-based study on the effects and consequences of OCD on SS. Moreover, we provide further information about OCD and SS, including cumulative incidence, stratification for risk factors, and interactions with comorbidities.

## 2. Materials and Methods

### 2.1. Database and Study Design

The study was a retrospective cohort study. We conducted a nationwide, population-based study using data from the National Health Insurance Research Database (NHIRD) in Taiwan. This database enrolled almost 99% of the 23 million beneficiaries in Taiwan, and includes all insurance claims, outpatient and emergency room visits, and hospitalizations. The NHIRD contains anonymized linked data from epidemiological studies, including patient demographics, healthcare services data, medications dispensed from general practitioners, community pharmacies, and hospitals, International Classification of Diseases 9th Revision Clinical Modification (ICD-9-CM) diagnostic codes, and drug codes. Several published epidemiological studies have applied the NHIRD extensively [15,16]. Of the 23 million, only 1 million subjects were sampled in this study, and their data were collected from 1999 to 2013. The test database was de-identified. The Institutional Review Board of Chung Shan Medical University Hospital approved this study.

### 2.2. Selection of Case and Control Groups

The study population included only patients with newly diagnosed obsessive-compulsive disorder (OCD; ICD-9-CM code = 300.3) from 2000 to 2012 who had three or more outpatient visits or more than one hospitalization, in order to ensure the accuracy of the diagnoses. The cohort index date was set as the date on which the patient was first diagnosed with OCD. To ensure the inclusion of new-onset subjects, we excluded those with a diagnosis of Sjögren’s syndrome (ICD-9-CM = 710.2) before the index date. The non-OCD group was defined as participants who were not diagnosed with Sjögren’s syndrome betwee 1999 and 2013. Baseline characteristics were age, sex, hypertension (ICD-9-CM = 401–405), hyperlipidemia (ICD-9-CM = 272.0–272.4), diabetes mellitus (ICD-9-CM = 250), obesity (ICD-9-CM = 278), coronary artery disease (ICD-9-CM = 410–414), stroke (ICD-9-CM = 430–438), and cancer (ICD-9-CM = 140–208). This study determined the comorbidities one year before the index date. Firstly, we used 1:10 matching by age and sex to determine an index date for the non-OCD group, in order to acquire the same starting point for both groups. We also performed a propensity score match by age, sex, hypertension, hyperlipidemia, diabetes mellitus, obesity, coronary artery disease, stroke, and cancer, in order to control for confounding factors between the two groups. A flowchart of the study is shown in Figure 1. The propensity score probability was estimated using logistic regression. The binary variables were the OCD and non-OCD groups. By matching the propensity scores, the heterogeneity of the baseline characteristics and comorbidities were balanced between the two groups.

### 2.3. Exposure Assessment

This study defined the outcome variable as a diagnosis of SS (ICD-9-CM = 710.2) with at least three outpatient visits or one hospitalization. For patients with SS, the review committee evaluated all submitted information according to the American European Consensus Group [17]. In Taiwan, the catastrophic illness certificate (CIC) is a requirement for additional benefits associated with a devastating illness. Suppose a patient wants to apply for the CIC. In that case, a rheumatologist must provide relevant clinical and laboratory information (including physical examination, anti-SS-A Ab examination, salivary gland biopsy, saliva scintillation imaging, etc.). Previous studies have also used CIC to define SS [18,19,20]. Therefore, as an additional criterion, we confirmed that subjects had developed SS on the basis of their having a CIC for SS. Subjects were followed up until the occurrence of SS, 31 December 2013, or their withdrawal from the national insurance system, whichever occurred first. To assess the robustness of our findings and reduce potential confounders, we conducted a sensitivity analysis to further examine the association between history of OCD and non-OCD and SS risk on the basis of diagnosis of SS with the usage of hydroxychloroquine, which is a common immuno-modulator in the treatment of SS, in one year. In this model, the ability to diagnose SS was restricted to rheumatologists.

### 2.4. Statistical Analysis

To compare the characteristics of the OCD and non-OCD groups, we utilized the chi-square test or Fisher’s exact test for categorical variables and Student’s t-test for continuous variables. We also used Kaplan–Meier analysis to calculate the cumulative incidence of SS and the log-rank test to test the significance. Furthermore, the Cox proportional hazard model was used to estimate the SS hazard ratio between the OCD and non-OCD groups after adjusting for age, sex, and comorbidities. We also performed a sensitivity analysis to examine the robustness of our findings by diagnosis of SS with usage of hydroxychloroquine in one year. The statistical software used was SPSS version 18.0 (SPSS Inc., Chicago, IL, USA).

## 3. Results

A total of 1678 patients were included in the OCD cohort before matching the propensity score. Similarly, 16,780 non-OCD patients were included in the control cohort. The OCD and control cohorts showed no significant differences with respect to their age and sex distributions (Table 1). Patients with OCD had a higher prevalence of pre-existing comorbidities such as hypertension, hyperlipidemia, diabetes mellitus, coronary artery disease, stroke, and cancer than non-OCD patients (*p* < 0.05). Finally, after propensity score matching, 1678 patients were included in the OCD cohort and 3356 non-OCD patients were included in the comparison cohort.

Table 2 illustrates the study population’s incidence density rate and adjusted hazard ratio (aHR) for SS. Patients with OCD and controls were followed up for 13,077 and 25,856 person-years, respectively. The overall SS incidence density rate was substantially higher in patients with OCD than in controls (1.9 vs. 0.6 per 1000 person-years). After adjusting for covariates, patients with OCD had a significantly higher risk of SS than non-OCD controls (adjusted HR = 3.41, 95% CI = 1.79–6.47). Those aged ≥65 years had the highest risk of SS (adjusted HR = 5.04, 95% CI = 1.10–23.12) in the full sample.

Table 3 demonstrates the sex- and age-stratified effects of OCD on SS. Overall, patients with OCD aged <65 years had a 3.97-fold (95% C.I. = 1.93–8.19) increased risk of SS compared to patients ≥65 years. Furthermore, the SS risk significantly increased over the 2-year follow-up period after OCD diagnosis (Table 4).

Figure 2 shows the Kaplan–Meier curve measuring a higher cumulative incidence of Sjögren’s syndrome in patients with OCD than in controls (log-rank test, *p* < 0.001).

After the sensitivity analysis, OCD patients had 5.73-fold greater risk of developing SS as compared with non-OCD patients (95% CI, 1.82 to 18.05) (Table 5).

## 4. Discussion

This is the first nationwide population-based study to assess the risk of developing SS in patients with OCD. The most important clinically relevant findings are as follows: (1) there is a 3.4-fold greater risk of SS in patients with OCD; (2) the risk of developing SS is significantly higher in women and the elderly in the full sample, and (3) there is a higher incidence of subsequent SS among patients with OCD over a follow-up duration of >2 years.

In the present study, females were revealed to have a higher risk than males of developing SS. This result corresponds to previous reports that SS occurs predominantly in women [21,22]. Sjögren’s syndrome is regarded as a disease occurring mainly in females, with a female/male ratio of 10:1 [21]. Importantly, our analysis indicated that the risk of SS in a two-year follow up was higher than that at baseline. OCD is usually associated with multiple comorbidities [23]. Therefore, physicians should be alert regarding the progression of these comorbidities. Careful reassessment during each patient visit is essential, especially among elderly women. Accordingly, even if a patient does not have comorbidities when OCD is diagnosed, incident systemic diseases may develop after the diagnosis and considerably increase the risk of SS.

Although SS mostly occurs in the fourth decade of life, our study indicates that an increased risk of developing SS affects patients in old age [24]. Differences in SS between age groups may have diverse presentations and outcomes [25,26]. The involvement of extraglandular manifestations in all ages has been hypothesized [27,28]. Previous findings suggest that psychiatric disorders are prevalent among patients with SS who may then demand psychiatric services and appropriate psychotropic therapy [29,30]. Our study afforded evidence supporting the idea that the risk of SS in patients with OCD may continuously increase. Some studies have provided comparative evidence that patients with SS had a higher prevalence of OCD than patients without SS [14]. Psychiatric involvement may be an early manifestation of SS, and is related to psychological, social, or economic consequences [6,31]. Sjögren’s syndrome primarily causes patients to suffer from a dry sensation, photophobia, burning, and fluctuating blurry vision, decreasing patients’ quality of life. Other significant related complications include arthritis, vasculitis, pulmonary fibrosis, thyroiditis, renal and hepatic involvements, and an increased risk of developing malignancy, especially non-Hodgkin’s lymphoma [32,33]. Therefore, early diagnosis and appropriate interventions are crucial for preventing the negative impacts of OCD in patients with SS.

There are several possible explanations for the link between SS and OCD. First, some OCD and immune-related comorbidities in families support the hypothesis of genetic susceptibility. Autoimmune diseases may trigger OCD presentation via shared genetic or epigenetic mechanisms [34,35]. Second, abnormalities in serotonergic signaling mediate crosstalk between the brain and the immune system, thus playing a significant role in OCD development and activation of autoimmune processes [36,37]. Third, this association with SS may be due to the fact that OCD patients are more medicalized (i.e., get more medical attention) than the non-OCD population.

Currently, diagnosis of SS is based on established classification criteria: the 2002 AECG Classification Criteria, 2012 SICCA criteria, 2016 ACR/EULAR criteria [17,38,39]. The majority of rheumatologists in Taiwan during study period applied the 2002 AECG criteria for the diagnosis of SS [40]. A primary SS diagnosis requires symptoms that meet four of the six AECG criteria or three of the four objective diagnostic criteria. A diagnosis of secondary SS requires the presence of a well-defined primary connective tissue disease, one subjective SS symptom, and symptoms meeting two of the three objective AECG diagnostic criteria for SS [17]. Although the present study is based on a recorded database, it is noteworthy for several reasons: First, it is a nationwide study which provides a sufficiently large sample size to avoid selection bias. Second, our study aimed to investigate this association using a population-based database with 1 million subjects and 14 years of cohort follow up. Therefore, the results are relatively comprehensive, with minimal recall bias. Third, we attempted to minimize the possibility of confounding factors by using propensity scores, stratification, and a multivariable regression model. We observed consistent results for the association between OCD and the subsequent development of SS in sensitivity analyses.

Although this study successfully demonstrates the crucial role of OCD in SS etiopathogenesis, it has certain limitations. First, diagnosis of OCD and occurrence of SS were determined on the basis of the diagnostic codes certified by physicians who cared for the patients [14,23]. However, the accuracy of OCD and SS diagnosis in Taiwan’s NHIRD has been validated [20,41,42]. Second, comprehensive information on laboratory data, family history, and personal lifestyle was lacking in the NHIRD registry database. Therefore, the clinical usefulness of biomarkers for disease activity assessment remains uncertain. Third, it is well-known that some psychiatric agents may induce or worsen SS. The lack of information regarding pharmacological therapies for OCD prior to SS diagnosis is a limitation of the study. Finally, our database is a population-based database from Taiwan. Although it still covers 99% of Taiwan’s population of 23 million, most of the population in this study is of Han ancestry; hence, any attempts to generalize these results may be influenced by the various ethnic backgrounds existing outside Taiwan. In addition, the patients with SS in the dataset were moderate to severe cases, and most of them were managed by the rheumatology department in medical centers. Therefore, the generalizability of these results to patients at other levels of hospitals or primary care settings, or other specialties, may be limited.

## 5. Conclusions

This nationwide population-based case–control cohort study revealed that OCD was correlated with a higher risk of incidental SS in a population-based. Clinicians should be aware that autoimmune symptoms are probable in patients with OCD, especially after a 2-year follow up. Clinically assessing psychiatric stress is important in managing patients with SS.

## Figures and Tables

**Figure 1 ijerph-18-05936-f001:**
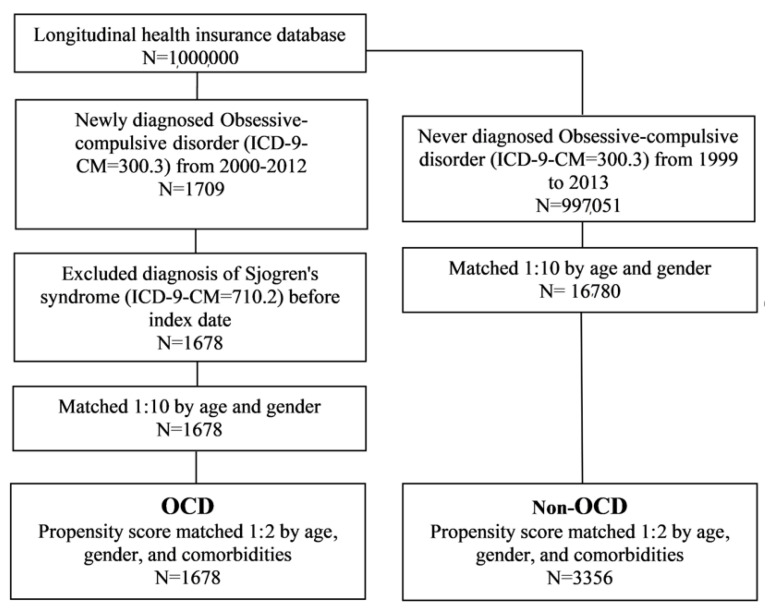
Flowchart of the study design. OCD: obsessive-compulsive disorder.

**Figure 2 ijerph-18-05936-f002:**
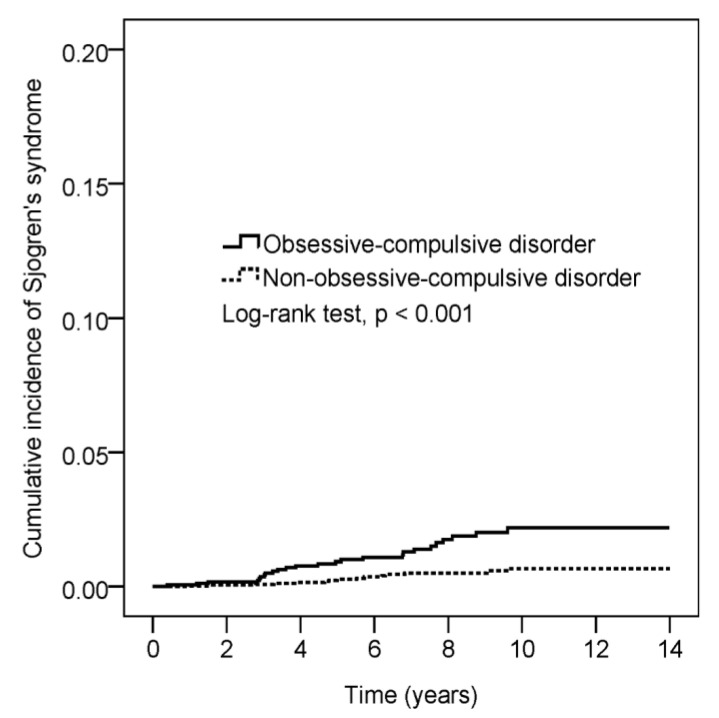
Cumulative incidence comparison of Sjörgen’s syndrome for patients with and without obsessive-compulsive disorder.

**Table 1 ijerph-18-05936-t001:** Demographic characteristics of obsessive-compulsive disorder and non-obsessive-compulsive disorder.

	Before Propensity Score Matching					After Propensity Score Matching				
	Obsessive-Compulsive Disorder (N = 1678)		Non-Obsessive-Compulsive Disorder (N = 16,780)		*p*-Value	Obsessive-Compulsive Disorder (N = 1678)		Non-Obsessive-Compulsive Disorder (N = 3356)		*p*-Value
	*n*	%	*n*	%		*n*	%	*n*	%	
Age					1					0.988
<20	329	19.6	3290	19.6		329	19.6	655	19.5	
20–39	735	43.8	7350	43.8		735	43.8	1458	43.4	
40–64	504	30.0	5040	30.0		504	30.0	1016	30.3	
≧65	110	6.6	1100	6.6		110	6.6	227	6.8	
Mean ± SD	35.6 ± 16.6		35.6 ± 16.6		1	35.6 ± 16.6		36.1 ± 17		0.332
Gender					1					0.889
Female	836	49.8	8360	49.8		836	49.8	1679	50.0	
Male	842	50.2	8420	50.2		842	50.2	1677	50.0	
Hypertension	186	11.1	1144	6.8	<0.001	186	11.1	378	11.3	0.850
Hyperlipidemia	97	5.8	436	2.6	<0.001	97	5.8	202	6.0	0.736
Diabetes mellitus	51	3.0	193	1.2	0.002	79	4.7	160	4.8	0.925
Obesity	2	0.1	11	0.1	0.334 ^†^	2	0.1	3	0.1	1 ^†^
Coronary artery disease	52	3.1	291	1.7	<0.001	52	3.1	107	3.2	0.864
Stroke	51	3.0	193	1.2	<0.001	51	3.0	92	2.7	0.549
Cancer	36	2.1	166	1.0	<0.001	36	2.1	75	2.2	0.839

^†^ Fisher’s exact test.

**Table 2 ijerph-18-05936-t002:** Cox proportional hazard model analysis for risk of Sjögren’s syndrome.

	No. of Sjögren’s Syndrome	Observed Person-Years	ID	Crude HR	95% C.I.	Adjusted HR ^†^	95% C.I.
Obsessive-compulsive disorder							
No	15	25,856	0.6	1		1	
Yes	25	13,077	1.9	3.31	1.74–6.28	3.41	1.79–6.47
Age							
<20	3	7702	0.4	1		1	
20–39	11	17,568	0.6	1.59	0.44–5.70	1.40	0.39–5.03
40–64	19	11,442	1.7	4.22	1.25–14.26	3.14	0.90–10.96
≥65	7	2221	3.2	8.14	2.10–31.48	5.04	1.10–23.12
Gender							
Female	29	19,016	1.5	1		1	
Male	11	19,918	0.6	0.36	0.18–0.73	0.41	0.20–0.83
Hypertension	10	3856	2.6	3.05	1.49–6.24	1.50	0.61–3.66
Hyperlipidemia	6	1884	3.2	3.58	1.50–8.55	2.72	0.99–7.46
Diabetes mellitus	1	1596	0.6	0.61	0.08–4.43	0.18	0.02–1.43
Coronary artery disease	3	1125	2.7	2.75	0.85–8.91	0.98	0.26–3.69
Stroke	2	890		2.29	0.55–9.49	0.78	0.18–3.45
Cancer	1	668	1.5	1.51	0.21–11.01	0.73	0.10–5.44

ID: incidence density (per 1000 person-years). ^†^ Adjusted for age, gender, hypertension, hyperlipidemia, diabetes mellitus, coronary artery disease, stroke, and cancer.

**Table 3 ijerph-18-05936-t003:** Subgroup analysis of the Cox proportional hazard model.

	Obsessive-Compulsive Disorder		Non-Obsessive-Compulsive Disorder			
	N	No. of Sjögren’s Syndrome	N	No. of Sjögren’s Syndrome	HR	95% CI
Age						
<65	1568	22	3129	11	3.97	1.93–8.19
≥65	110	3	227	4	1.46	0.33–6.55
*p* for interaction = 0.244						
Gender						
Female	836	17	1679	12	2.84	1.35–5.94
Male	842	8	1677	3	5.24	1.39–19.73
*p* for interaction = 0.426						

**Table 4 ijerph-18-05936-t004:** Subgroup analysis of Cox proportional hazard model.

	N	No. of Sjögren’s Syndrome	Crude HR	95% C.I.	Adjusted HR ^†^	95% C.I.
Follow-up duration ≤2 years						
Obsessive-compulsive disorder ^†^						
No	3356	2	1		1	
Yes	1678	3	2.99	0.5–17.88	3.03	0.51–18.16
Follow-up duration >2 years						
Obsessive-compulsive disorder ^‡^						
No	3284	13	1		1	
Yes	1649	22	3.36	1.69–6.67	3.50	1.76–6.96

^†^ Adjusted for age, gender, hypertension, coronary artery disease, and stroke. ^‡^ Adjusted for age, gender, hypertension, hyperlipidemia, diabetes mellitus, coronary artery disease, stroke, and cancer.

**Table 5 ijerph-18-05936-t005:** Sensitivity analysis using the usage of HCQ in patients with Sjögren’s syndrome.

	No. of Sjögren’s Syndrome	Observed Person-Years	ID	Crude HR	95% C.I.	Adjusted HR ^†^	95% C.I.
Obsessive-compulsive disorder							
No	4	25,906	0.2	1		1	
Yes	11	13,137	0.8	5.44	1.73–17.08	5.73	1.82–18.05

ID: incidence density (per 1000 person-years). ^†^ Adjusted for age, gender, hypertension, hyperlipidemia, diabetes mellitus, coronary artery disease, stroke, and cancer.

## Data Availability

All data relevant to the study are available from the National Health Insurance Research Database (NHIRD), which is provided by the National Health Insurance (NHI) administration, Ministry of Health and Welfare of Taiwan and the National Health Research Institutes (NHRI) of Taiwan. It is not publicly available because it restricts only the researchers or clinicians who applied and signed an agreement with NHRI are eligible to apply for the National Health Insurance Research Database (NHIRD). The following is the official website of the NHIRD (https://nhird.nhri.org.tw/).

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
