# Peer review of "Increased Risk of Sjögren’s Syndrome in Patients with Obsessive-Compulsive Disorder: A Nationwide Population-Based Cohort Study"

_ijerph, 2021, doi:10.3390/ijerph18115936_

Round 1
Reviewer 1 Report
Thank you for the opportunity to review this manuscript about this issue which gets very minimal attention in the literature and in clinical settings. My suggestions for improvement are below. Background- It would be helpful to provide more explanation about the link/overview of previous studies showing OCD as an initial symptom/preceding SS for readers who don't know anything about this area.
- To better understand the significance of this manuscript, and because the authors write that studying the impact of OCD on SS has "become of critical concern" it would be helpful to learn more about the percentage of people this might impact (no prevalence rates are provided for SS for example) - no mention of how many patients with OCD have autoimmune diseases, etc.
- The methods in this study are justified and well explained.
- For readers not familiar with this, please explain why a sensitivity analysis was performed including usage of hydroxychloroquine (this isn't explained until later in the discussion)
- Please specify in this first paragraph if the risk of developing SS is significantly higher in women and the elderly in those with OCD or in the full sample, same with the results about developing SS in patients in old age. Otherwise it is not clear if this was part of the aims of the study, or additional information learned.
- Please spell out which of these findings are consistent with those published in an earlier study to make easier for readers to understand
- It would be helpful to provide a discussion of generalizability - to different types of medical settings within Taiwan, outside of Taiwan, etc.
- It appears that this is an overstatement "clinicians should be aware that SS is probable in patients with OCD" probably for those with autoimmune symptoms seems more accurate than probable with all patients with OCD.
Author Response
We would like to thank the reviewer for your extensive assessment of our manuscript, and for important and helpful comments and suggestions. We have taken all the remarks into account, in a manner that is described in detail below together with our answers to certain comments. We think that, following the reviewers’ suggestions, our manuscript has gained in clarity and hope that the changes made will be considered satisfactory. The major changes are listed below:
Comment 1: It would be helpful to provide more explanation about the link/overview of previous studies showing OCD as an initial symptom/preceding SS for readers who don't know anything about this area.
Response: we had provided more explanation about the link showing OCD as an initial symptom/preceding SS on the background P2L1 and added references as the following:
Psychological abnormalities usually accompany chronic diseases. As previous studies have revealed, neuropsychiatric manifestations in SS can occur during its course and at the onset of the disease.
- Hammett, E. K.; Fernandez-Carbonell, C.; Crayne, C.; Boneparth, A.; Cron, R. Q.; Radhakrishna, S. M. Adolescent Sjogren's syndrome presenting as psychosis: a case series. Pediatr Rheumatol Online J. 2020, 18,15; DOI: 10.1186/s12969-020-0412-8.
- Ampélas, J. F.; Wattiaux, M. J.; Van Amerongen, A. P. Psychiatric manifestations of lupus erythematosus systemic and Sjogren's syndrome. Encephale. 2001, 27, 588-599.
- Chebli, S.; Zgueb, Y.; Ouali, U.; Taleb, S.; Nacef, F. Bipolar Disorder as Comorbidity with Sjögren's Syndrome: What Can We Do?. Case Rep Psychiatry. 2020, 8899615; DOI: 10.1155/2020/8899615.
Comment 2:To better understand the significance of this manuscript, and because the authors write that studying the impact of OCD on SS has "become of critical concern" it would be helpful to learn more about the percentage of people this might impact (no prevalence rates are provided for SS for example) - no mention of how many patients with OCD have autoimmune diseases, etc.
Response: We had provided the prevalence rates of the SS and autoimmune diseases in patients with OCD on the introduction section P1 and P2 L7as the following:
Sjögren syndrome (SS) is one of the most common autoimmune disease that occurs worldwide, and its prevalence is approximately 0.5 to 1.5% of the general population.
Over the last few years, studies have reported an increased prevalence of OCD in pa-tients with autoimmune diseases up to 17.8%,
Comment 3:The methods in this study are justified and well explained.
For readers not familiar with this, please explain why a sensitivity analysis was performed including usage of hydroxychloroquine (this isn't explained until later in the discussion)
Response: We added the sensitivity analysis in the method section P3L16as the following:
To assess the robustness of our finding and reduce the potential confounder, we conducted a sensitivity analysis to examine further the association between the history of OCD and non-OCD and SS risk by Moreover, we conducted a sensitivity analysis by the diagnosis of SS with the usage of hydroxychloroquine, which is a common immuno-modulator in the treatment of SS in one year. In this model, the diagnosis of SS was restricted to rheumatologists.
Comment 4:Please specify in this first paragraph if the risk of developing SS is significantly higher in women and the elderly in those with OCD or in the full sample, same with the results about developing SS in patients in old age. Otherwise it is not clear if this was part of the aims of the study, or additional information learned.
Response: We specify this on the result P4L23, and discussion section P6L17as the following:
Those aged ≥ 65 years had the highest risk of SS (adjusted HR = 5.04, 95% CI =1.10-23.12) in the full sample.
2) The risk of developing SS is significantly higher in women and the elderly in the full sample,
Comment 5:Please spell out which of these findings are consistent with those published in an earlier study to make easier for readers to understand
Response: We spell out findings consistent with those published in an earlier study in the discussion sectionP6L20 as the following:
In the present study, females and the elder have a higher risk than males and the younger developing SS. This result corresponds to previous reports of SS occurring predominantly in postmenopausal women. Sjögren’s syndrome has been regarded as a disease occurring mainly in females with a female/male ratio of 10:1.
Comment 6:It would be helpful to provide a discussion of generalizability - to different types of medical settings within Taiwan, outside of Taiwan, etc.
Response: We added a discussion of generalizability to different types of medical settings within Taiwan, outside of Taiwan on the discussion section P8L10 as the following:
Finally, our database is a population-based database from Taiwan. Although it still covers 99% of Taiwan’s 23 million population, most of the population in this study is of Han ancestry; hence, this result may be influenced by various ethnic backgrounds outside Taiwan. In addition, patients with SS in the dataset were moderate to severe cases, and most of them were managed by the rheumatology department in medical centers. Therefore, the generalizability of these results to patients in other levels of hospitals or primary care settings, or other specialties may be limited.
Comment 7:It appears that this is an overstatement "clinicians should be aware that SS is probable in patients with OCD" probably for those with autoimmune symptoms seems more accurate than probable with all patients with OCD.
Response: We corrected this overstatement on the conclusion by your suggestion as to the following:
Clinicians should be aware that autoimmune symptom is probable in patients with OCD, especially after 2 years of follow-up. Clinically assessing psychiatric stress is important in managing patients with SS.

Reviewer 2 Report
Dear Authors, thank you very much for your paper. You reviewed an important topic regarding SS, and surely your study provide additinal information and data collection on the topic. The paper is well written, scientific approach is good, data collection is strong and results are well presented.
As you know, several aspects are widely discussed in literature on SS. I suggest to improve the paper with a list (and relative references) of the main important SS diseases-related, complications, as well general and most recent diagnostic criteria. I believe that such data should never be lacking in a paper on SS.
Thank you again for your interersting paper.
Author Response
We would like to thank the reviewer for your extensive assessment of our manuscript, and for important and helpful comments and suggestions. We have taken all the remarks into account, in a manner that is described in detail below together with our answers to certain comments. We think that, following the reviewers’ suggestions, our manuscript has gained in clarity and hope that the changes made will be considered satisfactory. The major changes are listed below:
Comment 1:As you know, several aspects are widely discussed in literature on SS. I suggest to improve the paper with a list (and relative references) of the main important SS diseases-related, complications, as well general and most recent diagnostic criteria. I believe that such data should never be lacking in a paper on SS.
Response: We had added the main important SS diseases-related, complications, as well general and most recent diagnostic criteria on the discussion section P7L18, and added reference as the following:
Sjögren syndrome primarily makes patients suffer from dry sensation, photophobia, burning, and fluctuating blurry vision, decreasing patients’ life quality. Other signifi-cant SS related complications include arthritis, vasculitis, pulmonary fibrosis, thyroiditis, , renal and hepatic involvements, and increase the risk of developing malignancy, especially non-Hodgkin’s lymphoma.
The current diagnosis of SS was based on established classification criteria 2002 AECG Classification Criteria, 2012 SICCA criteria, 2016 ACR/EULAR criteria. The majority of rheumatologists in Taiwan in this study period applied the 2002 AECG criteria for the diagnosis of SS. A primary SS diagnosis requires symptoms that meet 4 of the 6 AECG criteria or 3 of the four objective diagnostic criteria. A diagnosis of secondary SS requires the presence of a well-defined primary connective tissue disease, one subjective SS symptom, and symptoms meeting 2 of the three objective AECG di-agnostic criteria for SS.

Reviewer 3 Report
Diagnosis of Sjögren’s syndrome (SS) should be based on established classification criteria (2002 American-European Consensus Group Classification Criteria, 2012 SICCA criteria, 2016 ACR/EULAR criteria…). The authors do not provide information on what kind of classification criteria for SS were applied to the study patients. The reviewer is afraid the diagnosis of SS was made by subjective judgements by attending physicians without enough differential diagnosis. It seems that the authors do not consider anti SS-A Ab and salivary gland biopsy as important items for the diagnosis of RA (page 6, line 37-38). However, the two items are included in the all the above classification criteria (required items in 2002 AECG criteria).
Responsiveness to hydroxychroloquine is not included in diagnostic items for SS. Therefore, the discussion on page line 40 to 43 is not understandable.
The study targeted patients with obsessive-compulsive disorder. Psychotropic drugs can be causes of sicca symptoms which resembles those of SS.
Considering these shortcomings, this research paper do not contain minimum requirements to be considered as a study on SS.
Author Response
We would like to thank the reviewer for your extensive assessment of our manuscript, and for important and helpful comments and suggestions. We have taken all the remarks into account, in a manner that is described in detail below together with our answers to certain comments. The major changes are listed below:
Comment 1:Diagnosis of Sjögren’s syndrome (SS) should be based on established classification criteria (2002 American-European Consensus Group Classification Criteria, 2012 SICCA criteria, 2016 ACR/EULAR criteria…). The authors do not provide information on what kind of classification criteria for SS were applied to the study patients. The reviewer is afraid the diagnosis of SS was made by subjective judgements by attending physicians without enough differential diagnosis. It seems that the authors do not consider anti SS-A Ab and salivary gland biopsy as important items for the diagnosis of RA (page 6, line 37-38). However, the two items are included in the all the above classification criteria (required items in 2002 AECG criteria).
Response: Thanks for this expert comment. We provide the information on what kind of classification criteria for SS were applied to the study patients on discussion section P7L32 and method section P3L6 as the following :
The current diagnosis of SS was based on established classification criteria 2002 AECG Classification Criteria, 2012 SICCA criteria, 2016 ACR/EULAR criteria [17,38,39]. The majority of rheumatologists in Taiwan in this study period applied the 2002 AECG criteria for the diagnosis of SS.
For patients with SS, the review committee evaluated all submitted information according to the American European Consensus Group[17]. In Taiwan, the catastrophic illness certificate (CIC) is a requirement for additional benefits associated with a devastating illness. Suppose a patient wants to apply for CIC. In that case, a rheumatologist must provide relevant clinical and laboratory information (including physical examination, anti-SS-A Ab examination, salivary gland biopsy, and saliva scintillation imaging, etc.)
Comment 2 :Responsiveness to hydroxychroloquine is not included in diagnostic items for SS. Therefore, the discussion on page line 40 to 43 is not understandable.
Response: we amended the unclear content in the discussion section. We write the correction on the discussion section as the following :
Although our present study is based on a recorded database, it is noteworthy for several reasons: First, this was a nationwide study; this study provided a sufficiently large sample size to avoid selection bias. Second, our study aimed to investigate this association using a population-based database with 1 million and 14 years cohort follow-up. Therefore, the results more relatively are comprehensive with minimal recall bias. Third, we attempted to minimize the possibility of confounding by a propensity score, stratification and a multivariable regression model. We observed consistent results on the association between OCD and the subsequent development of SS in sensitivity analyses.
Comment 3: The study targeted patients with obsessive-compulsive disorder. Psychotropic drugs can be causes of sicca symptoms which resembles those of SS.
Response: We agree to this comment and realize this association. Thus, in our study, the diagnosis of SS was not based on symptoms only but needs to be reviewed by rheumatologists based on the classification criteria, including serological findings. We also did a sensitivity test on the restriction of patients with the coding of SS plus the use of hydroxychloroquine.
We had described this on the limitation on the discussion section P8L7 as the following :
Third, it is well known that some psychiatric agents may induce or worsen SS. The lack of information regarding pharmacologic therapies for OCD before SS diagnosis is a limit of the study.
Comment 4: Considering these shortcomings, this research paper do not contain minimum requirements to be considered as a study on SS.
Response: We understand the concerns of the validity of SS. Thus, we had added more description in the METHODS and DISCUSSION. Our previous publications for the validity of SS were also provided in references.
Chen HH, Perng WT, Chiou JY, Wang YH, Huang JY, Wei JC. Risk of dementia among patients with Sjogren's syndrome: A nationwide population-based cohort study in Taiwan. Semin Arthritis Rheum (IF: 4.751; Q1). 2019 Apr;48(5):895-899.
Wu XF, Huang JY, Chiou JY, Chen HH, Wei JC, Dong LL.Increased risk of coronary heart disease among patients with primary Sjogren's syndrome: a nationwide population-based cohort study. Sci Rep (IF: 3.998; Q1). 2018 Feb 2;8(1):2209.
Gau SY, Leong PY, Lin CL, Tsou HK, Wei JC. Higher Risk for Sjogren's Syndrome in Patients With Fi-bromyalgia: A Nationwide Population-Based Cohort Study. Front Immunol (IF: 5.085; Q1). 2021 Apr 12;12:640618. doi: 10.3389/fimmu.2021.640618.
We think that, following the reviewers’ suggestions, our manuscript has gained clarity and hope that the changes made will be considered satisfactory.

Round 2
Reviewer 3 Report
The authors responded the concerns the reviewer raised quickly.
However, numerical grammatical errors are noticed in the additional sentences in the revised manuscript. The authors should ask professional language-editing service before resubmission, the statement of which should be included in Acknowledgements.
Ref. 10 is missing.
Author Response
Thank the reviewer for your extensive assessment of our manuscript and important and helpful comments and suggestions. We have taken all the remarks into account in a manner described in detail below, together with our answers to certain comments. The major changes are listed below:
Comment 1: However, numerical grammatical errors are noticed in the additional sentences in the revised manuscript. The authors should ask professional language-editing service before resubmission, the statement of which should be included in Acknowledgements.
Response: We asked the professional language-editing service and described the statement in Acknowledgements in P9 as the following:
We appreciate MDPI for English editing providing a professional language editing service for our grammatical errors.
Comment 2: Ref. 10 is missing.
Response: We define the loss in Ref.10 as the following:
- Chebli, S.; Zgueb, Y.; Ouali, U.; Taleb, S.; Nacef, F. Bipolar Disorder as Comorbidity with Sjögren’s Syndrome: What Can We Do? Case Rep. Psychiatry 2020, 8899615, doi:10.1155/2020/8899615.
Following the reviewers’ suggestions, we think that our manuscript has gained clarity and hope that the changes made will be considered satisfactory.

Round 3
Reviewer 3 Report
(There are no comments)